# Maternal exposure to sulfonamides and adverse pregnancy outcomes: A systematic review and meta-analysis

Peixuan Li[1,2,3,4], Xiaoyun Qin[1,2,3,4], Fangbiao Tao[1,2,3,4], Kun Huang[1,2,3,4,5]*

**1** Department of Maternal, Child and Adolescent Health, School of Public Health, Anhui Medical University, Hefei, China, **2** Key Laboratory of Population Health Across Life Cycle (AHMU), MOE, Hefei, China, **3** NHC Key Laboratory of Study on Abnormal Gametes and Reproductive Tract, Hefei, China, **4** Anhui Provincial Key Laboratory of Population Health and Aristogenics, Hefei, China, **5** Scientific Research Center in Preventive Medicine, School of Public Health, Anhui Medical University, Anhui Province, Hefei, China

\* ahmuhuangk@163.com

**Data Availability Statement:** All relevant data are within the paper and its Supporting Information files.

**Funding:** The authors received no specific funding for this work.

## Abstract

### Background

Sulfonamides are widely used to treat infectious diseases during pregnancy. However, the safety of maternal exposure to sulfonamides is controversial. This study aims to systematically review the available studies and examine the effect of maternal sulfonamides use on adverse pregnancy outcomes.

### Methods

We searched PubMed, Science Direct, Web of Science, ClinicalTrials.gov, CNKI and Wanfang Database (in Chinese). The meta-analysis used random effects model or fixed effects model to obtain the total odds ratio (*OR*) for each outcome through Stata11.0 software. Study on the relationship between sulfonamide exposure during pregnancy and adverse pregnancy outcomes. The study design covered randomized controlled trials, cohort studies and case-control studies. The study protocol was registered in PROSPERO with protocol number CRD42020178687.

### Results

A total of 10 studies, and 1096350 participants were included for systematic review. Maternal exposure to sulfonamides was found to be possibly associated with increased risk of congenital malformations (*OR* = 1.21, 95% *CI* 1.07–1.37). The use of sulfonamides in the first trimester of pregnancy and during the entire pregnancy might be associated with congenital malformations.

### Conclusions

Maternal exposure to sulfonamides may be associated with offspring's congenital malformations. Prescription of sulfonamides for pregnant women is suggested to be carefully censored.

**Competing interests:** The authors have declared that no competing interests exist.

## Introduction

Sulfonamides are a very important class of drugs, with antibacterial, diuretic, hypoglycemic, antithyroid activity and other pharmacological effects [1]. They are usually used as human medicines, agriculture, aquaculture and animal husbandry [2]. People can be exposed to sulfonamides in every aspect of daily life. Studies have shown that sulfonamides were detected in drinking water, air, dust, soil vegetables and grains, and some animalistic food like meat, egg, milk, etc [3].

Sulfonamides are also the first drugs to be systematically used to prevent and treat human bacterial infections [4]. Because infections during pregnancy often cause serious maternal and fetal complications, sulfonamides have also been used as first-line agents in the second and third trimesters to treat and prevent urinary tract infections and other infections caused by susceptible microorganisms. The most commonly used sulfonamides during pregnancy is trimethoprim-sulfamethoxazole (TMP-SMX) [5, 6]. The American College of Obstetricians and Gynecologists also advised that sulfonamides could be used in the first trimester of pregnancy when other antibiotics are not available [7, 8].

Due to the widespread use of sulfonamides, the amount of sulfonamides exposed during pregnancy is also considerable. A large German study reported that the rate of TMP-SMX use in early pregnancy was 0.54% [9]. Data from the National Ambulatory Medical Care Survey (NAMCS) and National Hospital Ambulatory Medical Care Survey (NHMACS) had shown that 22% of pregnant women with uncomplicated urinary tract infections used sulfonamides during 2002–2011 [10]. In recent years, a study investigated 536 pregnant women aged 16–42 years from two geographically different study sites in Eastern China in 2015. Urinary antibiotics were overall detected in 41.6% of pregnant women, of which, the detection rate of sulfonamides antibiotics in urine was 13.6% [11]. A human bio-monitoring data including 369 pregnant women showed that sulfonamides were detected in 0.8% of newborn meconium, and the maximum level was found to be 75.7μg/kg [12].

The Food and Drug Administration (FDA) classifies a drug into five categories: A, B, C, D, and X [13]. Sulfonamides belong to the FDA' s Class C drugs, that is, animal studies have shown adverse effects on the fetus, and evidence from human studies is not sufficient [14, 15]. Currently, sulfonamides are not listed as prohibited drugs and are highly used during pregnancy. The potential effect of sulfonamides exposure during pregnancy on adverse pregnancy outcomes is inconclusive. Current evidence has shown that sulfonamides were more likely to cause adverse pregnancy outcomes than other antibacterial drugs [16–19]. There are also some controversies about the safety of using sulfonamides during pregnancy. However, to the best of our knowledge, there has been no integrated study to examine the impact of maternal sulfonamides exposure on pregnancy outcomes until now. Whether maternal exposure to sulfonamides will lead to adverse pregnancy outcomes, what kind of adverse pregnancy outcomes will be caused by sulfonamides exposure during pregnancy, and during which key gestational period will maternal sulfonamides exposure result in adverse pregnancy outcomes are issues worthy of research. Therefore, this study aims to provide an up-to-date systematic review and meta-analysis on whether maternal exposure to sulfonamides during pregnancy is associated with major adverse pregnancy outcomes, and to provide valuable information on the safety of sulfonamides usage as well as the rationale of sulfonamides prescription during pregnancy.

## Methods

This meta-analysis was evolved according to the recommended Preferred Reporting Items for Systematic Reviews and Meta-Analyses guideline [20].

## Search strategy

We searched PubMed, Science Direct, Web of Science, ClinicalTrials.gov, CNKI (China National Knowledge Infrastructure) and Wanfang Database (in Chinese). We used the following retrieval formula in Pubmed: (pregnancy OR pregnant OR conception OR fetation OR gestation) AND (sulfonamides OR sulfamethoxazole OR sulfametoxydiazine OR sulfaquinoxaline OR sulfamethazine OR sulfadiazine OR sulfasalazine OR sulfamethizole OR cotrimoxazole), there are no other special restrictions. We determined the types of adverse pregnancy outcomes based on the findings of the articles. The main adverse pregnancy outcomes for our meta-analysis were congenital malformations (including congenital malformations of the circulatory system, congenital malformations of the musculoskeletal system and Cleft lip ± palate), and the secondary adverse outcomes were spontaneous abortion, preterm birth/low birth weight.

## Study selection and quality evaluation

The criteria for eligible articles were as follows: (1) There was information on maternal exposure to sulfonamides and adverse pregnancy outcomes in titles and/or abstracts. (2) The study design covered case-control studies, cohort studies and randomized controlled trial. We searched articles in both Chinese and English database but finally did not find any articles that met the criteria in the Chinese database. After the retrieval, all articles were firstly imported into Note Express and duplicate literature was removed. Then, the literature was initially screened by overviewing the titles and abstracts, articles unrelated to the subject were excluded, and full-text studies were retained. Finally, for further reading, animal experiments, duplicate patient data-set, case reports and studies with no interested outcomes were excluded. This process was carried out independently by two researchers, and different opinions were reached an agreement according to the discussion. Two researchers independently extracted literature information. The information we extracted from the included literature mainly included first author, country, publication year, research type, outcome indicators, number of cases(exposed)/controls(unexposed) and total score of NOS (Jadad), as shown in Table 1.

According to the Cochrane guidelines, the quality of the included studies and the risk of bias were measured by the Newcastle-Ottawa Scale (NOS) [21] and Jadad Score [22]. The NOS scale was used to evaluate observational studies and the Jadad Score was used to evaluate randomized controlled trials, and all included literature was independently assessed by two reviewers. The NOS was based on the three sub scales, selection, comparability and outcome/exposure. The total NOS score ranged from 0 to 9, a score of 7 or higher was regarded as high quality. The total Jadad score ranged from 1 to 5, a score of 3 or higher was regarded as high quality. The quality assessment of all included studies was presented in Table 1. GRADEprofiler software was used to evaluate the GRADE level of evidence for each outcome [23]. The level of evidence is divided into four levels: high, moderate, low and very low. Observational studies were initially defined as low quality. The level of evidence will be reduced due to the following five factors: (1) risk of bias; (2) inconsistency; (3) indirectness; (4) imprecision; (5) publication bias. The following three factors may increase the quality level of evidence: (1) large effect; (2) dose response; (3) all plausible residual confounding.

## Data analysis

Adverse pregnancy outcomes in our study included: (1) Congenital malformations were the main outcome we were interested in this study, which indicated structural developmental abnormalities that occurred at birth. We extracted the first author, publication year and dichotomous data of included literature, imported them into Stata11.0 software, then we got

**Table 1. The basic characteristics and effect values of the included studies.**

| Number | Author | Year of publication | Country | Study period | Study design | Pregnancy outcomes | Number of cases (exposed)/controls (unexposed) | Total score of NOS (Jadad) |
|---|---|---|---|---|---|---|---|---|
| 1 | Yang J | 2011 | Canada | 1997–2000 | a retrospective cohort study | Preterm birth Low birth weight | 447/14537 | 7 |
| 2 | Czeizel AE | 2004 | Hungary | 1980–1996 | a case–control study | congenital malformations | 22843/38151 | 9 |
| 3 | Prasad MH | 1996 | India | - | a cohort study | spontaneous abortion preterm birth congenital malformations | 564/636 | 6 |
| 4 | Ratanajamit C | 2003 | Denmark | 1991–2001 | cohort study case–control study | congenital malformations Low birth weight Preterm birth spontaneous abortion | 3484/60175 3347/22599 | 7 |
| 5 | Muanda FT | 2018 | Canada | 1998–2009 | a nested case–control study | spontaneous abortion | 6612/65613 | 7 |
| 6 | Hansen C | 2016 | the United States | 2001–2008 | a cohort study | congenital malformations | 6688/6688 | 7 |
| 7 | Crider KS | 2009 | the United States | 1997–2003 | a case-control study | congenital malformations | 13155/4941 | 8 |
| 8 | Hill L | 1988 | United Kingdom | 1983 | a case-control study | congenital malformations | 676/676 | 6 |
| 9 | Brumfitt W | 1973 | England | 1973 | a randomized controlled trial | congenital malformations | 120/66 | 3 |
| 10 | Damkier P | 2019 | Denmark | 2000–2015 | a cohort study | congenital malformations | 22684/801648 | 8 |

forest plot with pooled odds ratios. In addition, we performed two subgroup analysis on the association between maternal exposure to sulfonamides and the major congenital malformations respectively by different gestational ages and by different types of malformations. The different gestational ages were divided into pre-pregnancy and first trimester of pregnancy and during the entire pregnancy. The different types of malformations were classified into 3 different types, including congenital malformations of the circulatory system, congenital malformations of the musculoskeletal system and Cleft lip±palate; (2) Spontaneous abortion. It was defined as a pregnancy with a diagnosis or procedure before the 20th week of gestation (ICD-10 codes O01-O03); (3) Preterm birth. It was defined as a gestational age below 37 completed weeks at birth; (4) Low birth weight, which was defined as newborn's birth weight less than 2500g. Due to the limited numbers of articles on the outcome of spontaneous abortion, preterm birth/low birth weight, we had not conducted subgroup analysis by gestational week.

Dichotomized variables were extracted and imported into Stata11.0 software, and these variables were expressed by odds ratio (*OR*) and 95% confidence interval (*CI*). Forest plot was used to present the pooled OR value. In the meta-analysis, $I^2$ was used as an indicator to quantify the degree of heterogeneity. We used $I^2$ to evaluate the statistical heterogeneity between studies. If $I^2 \leq 50\%$, and the heterogeneity was low and acceptable, then fixed effect model was used for meta-analysis; if $I^2 > 50\%$ and heterogeneity between studies was high, random effect model was adopted [24, 25]. We would also conduct further research on the included articles through subgroup analysis or sensitivity analysis [26]. Publication bias was detected by the funnel plot and Egger's/Begg's test. Sensitivity analysis was introduced to assess the effect of each individual study on the overall meta-analysis summary estimate. We also used sensitivity analysis to investigate the stability of the outcome of meta-analysis. We observed the pooled odds ratios after removing each study in turn, if the pooled odds ratios exceeded the original confidence interval, the outcome of meta-analysis was unstable; otherwise, the outcome of meta-analysis was regarded to be stable.

## Protocol registration

The protocol for this systematic review and meta-analysis has been registered in the PROS-PERO (Protocol No. CRD42020178687).

# Results

## Characteristics of the selected studies

From the 12118 retrieved studies, 10 studies matching the inclusion criteria were selected for meta-analysis, including 4 case-control studies, 4 cohort studies, 1 randomized controlled trial and 1 including both case-control and cohort studies. A total of 1096350 participants were included in the recruited 10 studies. The studies selection process was shown in Fig 1.

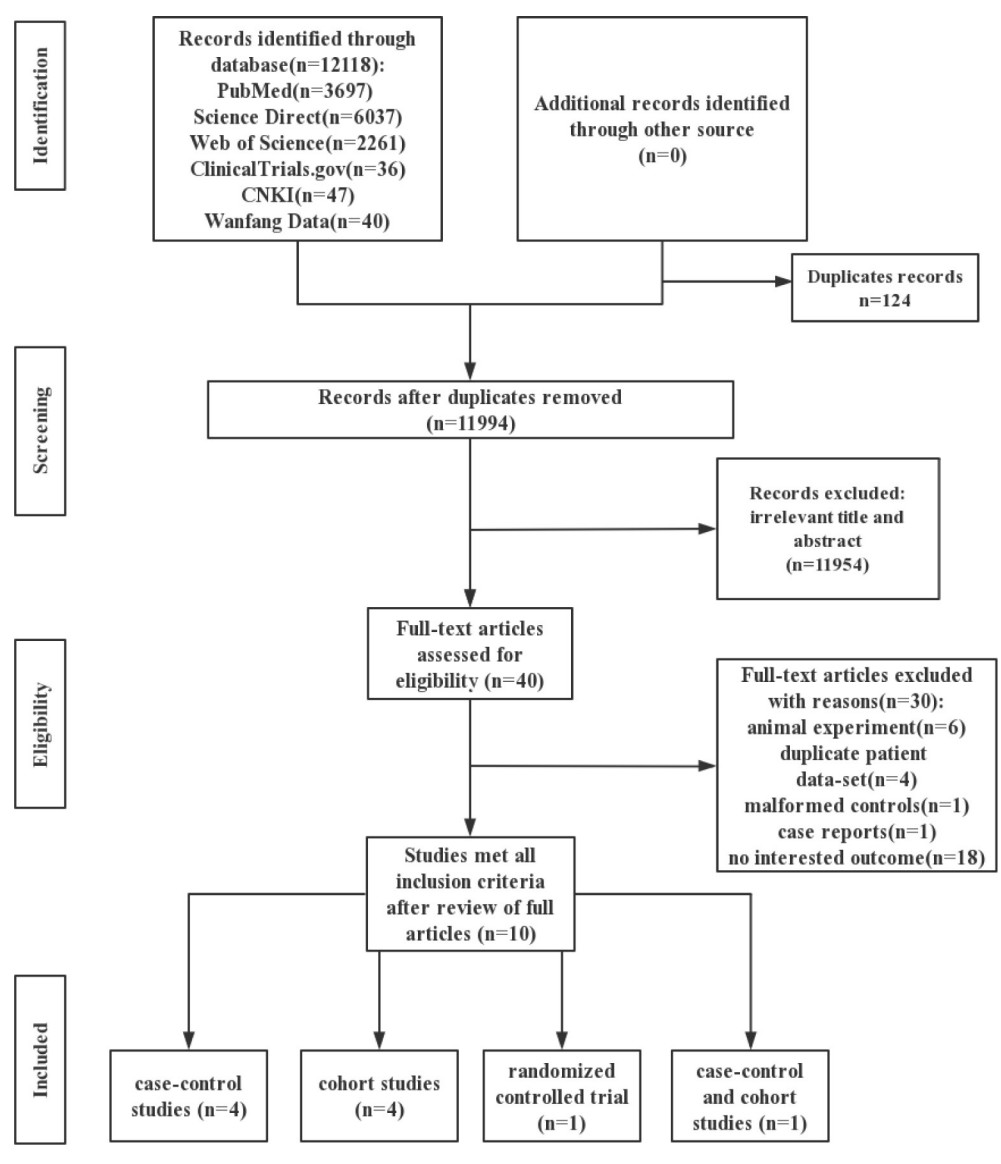

**Fig 1. Flow diagram of publications screening.**

## Maternal exposure to sulfonamides and congenital malformations (CM)

Eight studies were selected to analyze the association between maternal exposure to sulfonamides and CM [4, 18, 19, 27–31]. The combined analysis of 8 studies showed that the heterogeneity was 53.4% ($P = 0.036$), thus the random effects model was adopted. The results showed that maternal exposure to sulfonamides was associated with a high risk of congenital malformations ($OR = 1.21$, 95% $CI$ 1.07–1.37). Publication bias was detected by the funnel plot (S1 Fig) and the Egger's test ($t = 2.82$, $P = 0.030$, 95% $CI$ 0.19–2.71), and it indicated that there was publication bias. The sensitivity analysis results showed that the pooled odds ratios did not exceed the original confidence interval after removing each study in turn, and indicated that the result was stable (Fig 2).

Two subgroup analyses were further conducted. Firstly, we had performed the analysis on maternal exposure to sulfonamides by different gestational ages according to the available data. The periods of "pre-pregnancy and first trimester of pregnancy" and "during the entire pregnancy" were defined. The period of "pre-pregnancy and first trimester of pregnancy" referred to "1 month before pregnancy till the end of the first trimester". The period of "during the entire pregnancy" referred to "at any time during pregnancy". The results showed that maternal exposure to sulfonamides in various periods of gestation were associated with congenital malformations, overall $OR$ (95% $CI$) being 1.20(1.08–1.34). Maternal exposure to sulfonamides in pre-pregnancy and first trimester of pregnancy and during the entire pregnancy was both associated with congenital malformations, and the $OR$ (95% $CI$) was 1.18(1.04–1.33), and 1.26(1.01–1.57), respectively (Fig 3).

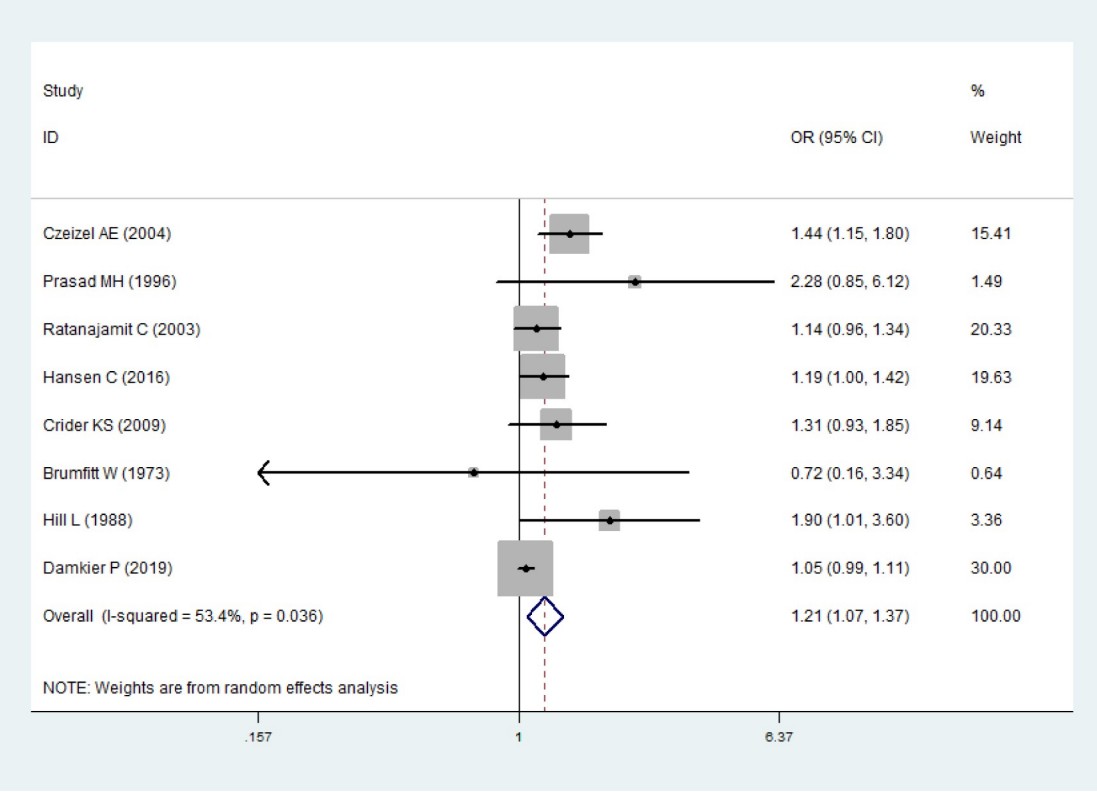

**Fig 2. Forest plot with pooled odds ratios of the effect of maternal exposure to sulfonamides on offspring's congenital malformations.**

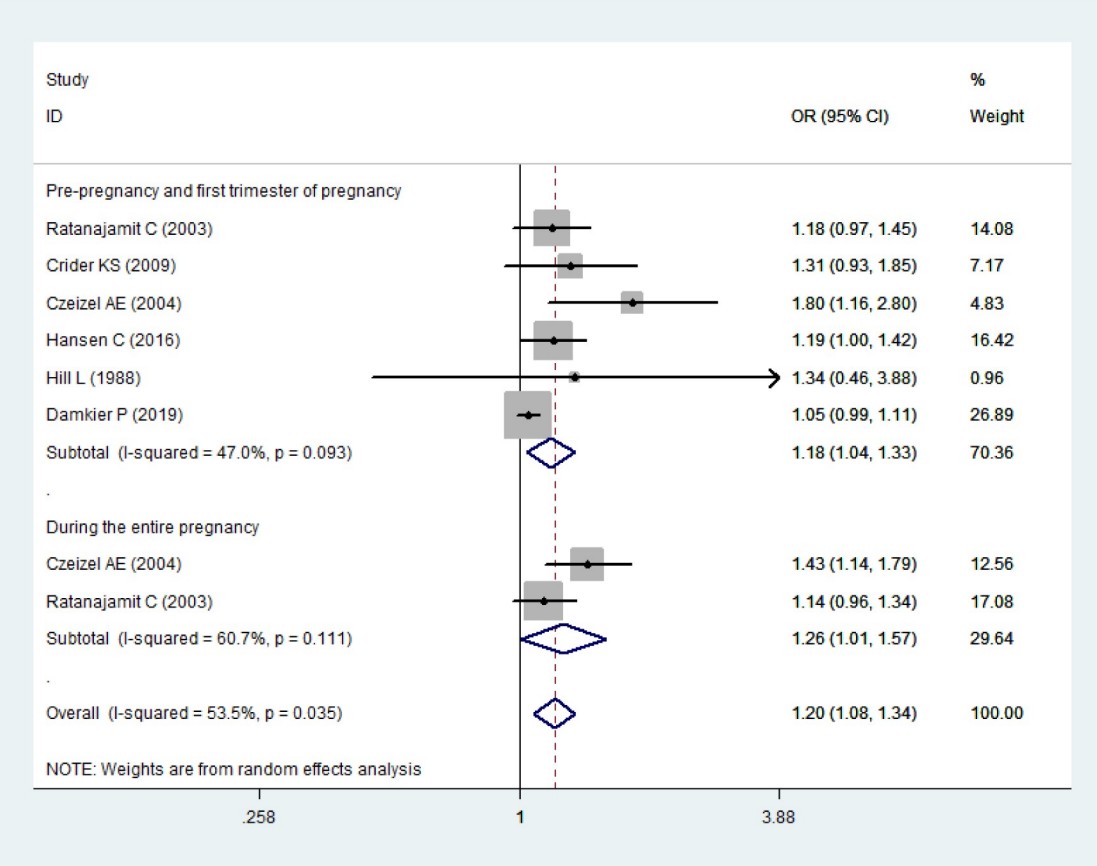

**Fig 3. Forest plot with pooled odds ratios of the effect of maternal exposure to sulfonamides on offspring's congenital malformations-subgroup analysis by periods of gestation.**

Secondly, we performed the analysis on the relationship between maternal exposure to sulfonamides and different types of congenital malformations. We extracted the dichotomized data from 3 articles for pooled analysis, as shown in Fig 4. It showed that maternal exposure to sulfonamides was associated with CM of the circulatory system ($OR$ = 1.34, 95% $CI$ 1.07–1.68) and the musculoskeletal system ($OR$ = 1.54, 95% $CI$ 1.10–2.16).

## Maternal exposure to sulfonamides and spontaneous abortion

Three studies were included for the analysis [4, 16, 29]. The heterogeneity was higher than 50% ($I^2$ = 91.4%, $P<0.05$), so the random effects model was adopted. There was no significant association between maternal exposure to sulfonamides and spontaneous abortion ($OR$ = 1.63, 95% $CI$ 0.91–2.91). Funnel plot (S2 Fig) and Egger's test ($t$ = 3.24, $P$ = 0.191, 95% $CI$ -29.75–50.07) was used to detect publication bias, and it indicated that publication bias was relatively small. It had shown from the sensitivity analysis that the outcome of meta-analysis was stable (Fig 5).

## Maternal exposure to sulfonamides and preterm birth/low birth weight

Three studies were included for the analysis [4, 17, 29]. No significant heterogeneity was found ($I^2$ = 83.9%, $P$ = 0.002), the random effects model was thus adopted. It showed that maternal exposure to sulfonamides was possibly not associated with preterm birth ($OR$ = 1.38, 95% $CI$

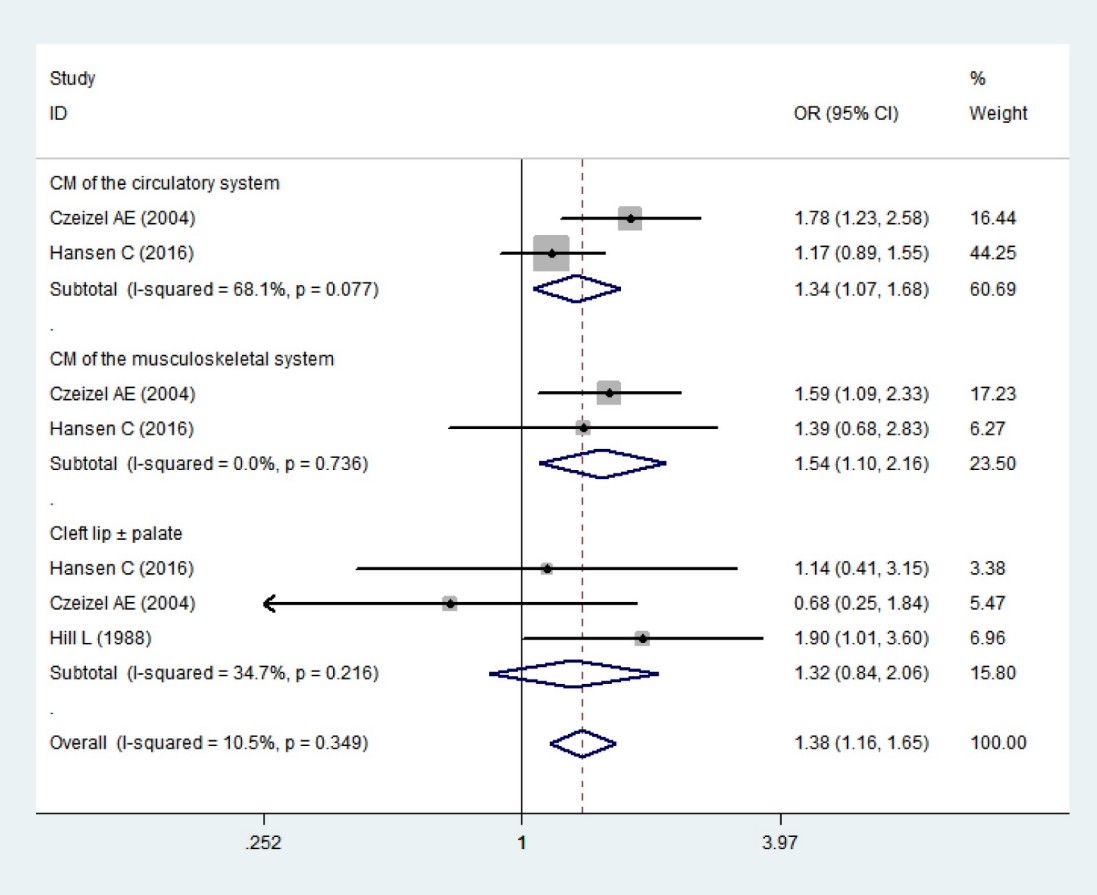

**Fig 4. Forest plot with pooled odds ratios of the effect of maternal exposure to sulfonamides on offspring's congenital malformations-subgroup analysis by types of malformations.**

0.94–2.03). Funnel plot (S3 Fig) and Egger' s test ($t = 1.10$, $P = 0.470$, 95% CI -35.48–42.19) was used to detect publication bias, and it indicated that publication bias was relatively small. It had shown from the sensitivity analysis that the outcome of meta-analysis was stable (Fig 6).

## GRADE quality of evidence

Three outcomes, congenital malformations, spontaneous abortion, preterm birth/low birth weight were presented in the current study. The level of evidence for each outcome was relatively low. The GRADE system evidence level and the reasons for promotion and demotion for each outcome were shown in Table 2.

## Discussion

This meta-analysis revealed that maternal exposure to sulfonamides at different perinatal time (pre-pregnancy, first trimester of pregnancy and during the entire pregnancy) might increase the risk of offspring' s congenital malformations. As far as we know, this is the first systematic review and meta-analysis that focuses on the association between maternal exposure to sulfonamides and adverse pregnancy outcomes.

In analysis on the relationship between maternal exposure to sulfonamides and fetal congenital malformations, in order to better distinguish the time of sulfonamides exposure prior

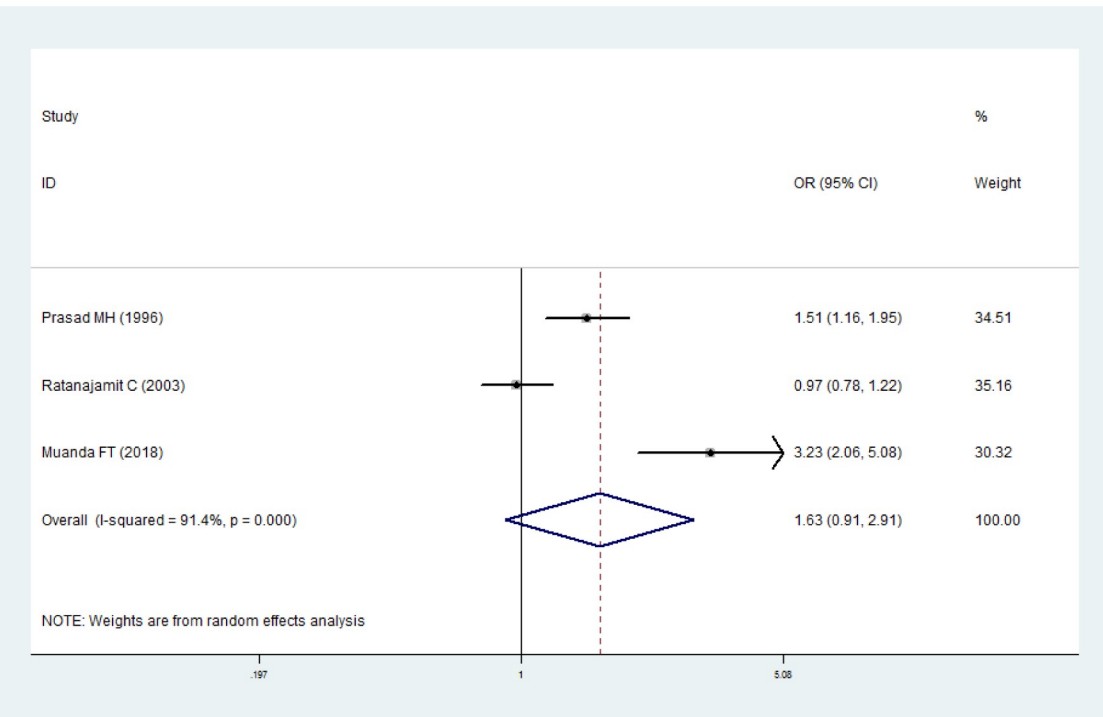

**Fig 5. Forest plot with pooled odds ratios of the effect of maternal exposure to sulfonamides on spontaneous abortion.**

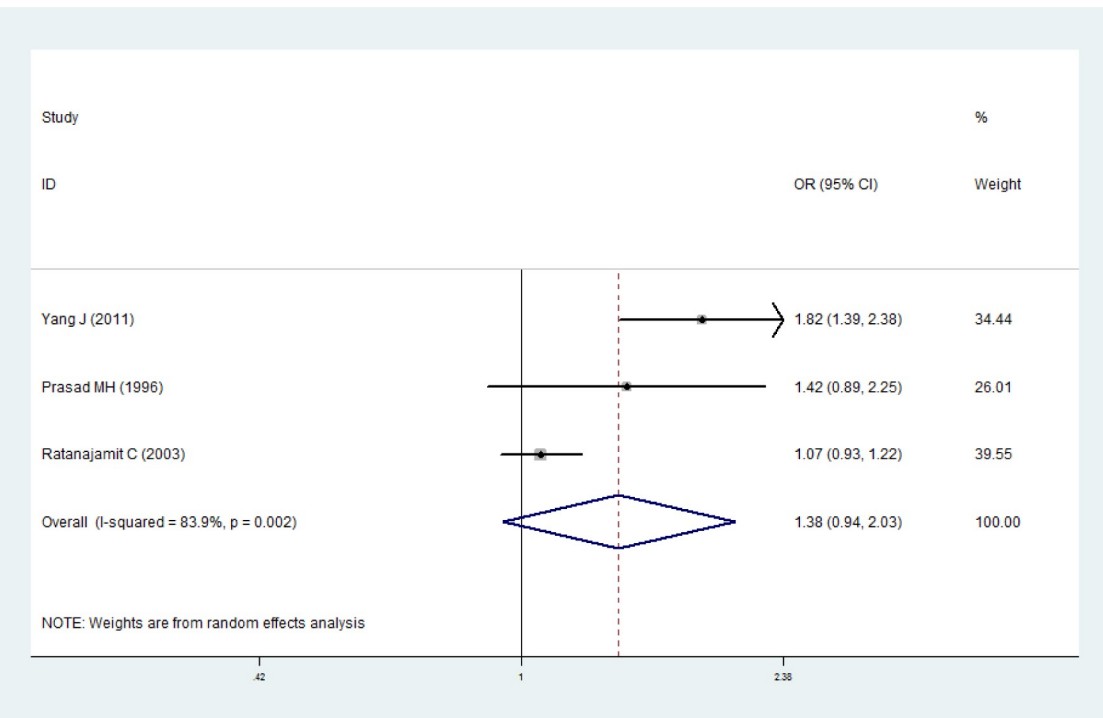

**Fig 6. Forest plot with pooled odds ratios of the effect of maternal exposure to sulfonamides on preterm birth/low birth weight.**

**Table 2. A summary table based on GRADE regarding the level of evidence for each outcome.**

| No of studies | Quality assessment | | | | | | No of patients | | Effect | | Quality | Importance |
|---|---|---|---|---|---|---|---|---|---|---|---|---|
| | Design | Risk of bias | Inconsistency | Indirectness | Imprecision | Other considerations | Sulfonamides | Control | Relative (95% CI) | Absolute | | |
| **congenital malformations (follow-up mean 1 years; assessed with: structural developmental abnormalities that occur at birth.)** | | | | | | | | | | | | |
| 8 | observational studies[1] | no serious risk of bias | no serious inconsistency | no serious indirectness | no serious imprecision | reporting bias[2] | 36674 cases 43768 controls and 1708/33540 exposed 44680/869213 unexposed | | OR 1.21 (1.07 to 1.37) | - | VERY LOW | CRITICAL |
| | | | | | | | | 4.9% | | 10 more per 1000 (from 3 more to 17 more) | | |
| **spontaneous abortion (follow-up mean 1 years; assessed with: a pregnancy with a diagnosis or procedure before the 20th week of gestation (ICD-10 codes O01-O03))** | | | | | | | | | | | | |
| 3 | observational studies[1] | no serious risk of bias | serious[3] | no serious indirectness | serious[4] | none | 9959 cases 88212 controls and 166/564 exposed 138/636 unexposed | | OR 1.63 (0.91 to 2.91) | - | VERY LOW | CRITICAL |
| | | | | | | | | 10% | | 53 more per 1000 (from 8 fewer to 144 more) | | |
| **preterm birth / low birth weight (follow-up mean 1 years; assessed with: a gestational age below 37 completed weeks at birth; newborn's birth weight less than 2500g)** | | | | | | | | | | | | |
| 3 | observational studies | no serious risk of bias | serious[5] | no serious indirectness | serious[4] | none | 350/4495 (7.8%) | 5204/ 75348 (6.9%) | OR 1.38 (0.94 to 2.03) | 24 more per 1000 (from 4 fewer to 62 more) | VERY LOW | IMPORTANT |

1 case-control and other study designs together

2 Egger's test (t = 2.82, P = 0.030, 95% CI 0.19–2.71)

3 $I^2$ = 91.4%, P<0.05

4 95% confidence interval of the pooled effect includes both 1) no effect and 2) appreciable benefit or appreciable harm

5 $I^2$ = 83.9%, P<0.05

to pregnancy and in first trimester of pregnancy, in the only study that separately described the use of sulfonamides before pregnancy, we found that maternal exposure to sulfonamides before pregnancy was associated with congenital malformations (*OR* = 2.4, 95% *CI* 1.1–5.3). As for the association between exposure to sulfonamides during pregnancy and offspring's congenital malformations, in all recruited studies, only one randomized controlled experiment was found with regards to the teratogenicity of using TMP-SMZ during pregnancy. It had not shown strong evidence that TMP-SMZ could cause serious teratogenic risks, which was inconsistent with the results of our study. Due to the small sample size and non-representative participants in the randomized controlled trial more experimental studies are needed to provide higher level evidence in the teratogenic effect of maternal exposure to sulfonamides.

Preterm birth is usually related to low birth weight. In our study, there were only three included articles concerning maternal exposure to sulfonamides and preterm birth/low birth weight. Among them, Ratanajamit C's study clearly indicated that low birth weight was restricted to full-term deliveries, while in Yang J's study, low birth weights covered preterm birth, and vice versa, and Prasad MH's study just included preterm birth without mentioning low birth weight. As it was difficult to clearly identify isolated preterm birth or low birth weight, we combined preterm birth and low birth weight as one outcome in analysis and did not found significant effect of maternal exposure to sulfonamides on preterm birth/low birth weight. We believe that more research data is needed to identify the relationship between maternal exposure of sulfonamides and preterm birth/low birth weight.

Some potential mechanisms might explain our findings that maternal exposure to sulfonamides was likely to increase the risk of fetal congenital malformations. Studies had shown that sulfonamides could cross the placental barrier [32]. Experimental studies revealed that sulfonamides had obvious teratogenic effects on mammalian animals [33, 34]. Folate is important for fetal development as it plays a key role in DNA synthesis during the rapid division of fetal cells [35]. Sulfonamides can inhibit cell proliferation by affecting folic acid synthesis. The process of folic acid synthesis is that dihydropteridine pyrophosphate and *p*-amino benzoic acid are condensed by dihydropteroate synthase to form dihydropterin, and glutamate is added to dihydropteroate by dihydrofolate synthase to form dihydrofolate, and then converts dihydrofolate to tetrahydrofolate by dihydrofolate reductase. Tetrahydrofolate is necessary for the synthesis of DNA and purines. The structure of sulfonamides is similar to *p*-amino benzoic acid, and compete with this compound for the biosynthesis of dihydropteroate, thereby reducing the synthesis of dihydrofolic acid [36, 37]. TMP-SMX, as the most commonly used sulfonamides during pregnancy, is a folic acid antagonist [14]. It may cross the placenta, consume folic acid and act on trophoblast cells to affect folate metabolism, thereby inhibit DNA synthesis [38, 39]. Further studies have suggested that pre-pregnancy and the first trimester of pregnancy may be a critical period for most organ and phylogenetic development, and exposure to sulfonamides during this period may lead to a higher risk of congenital malformations [40]. Since most of the studies we included were observationally designed, confounding by indication was crucial to explain the results. The use of sulfonamides during pregnancy was mostly aimed to treat infectious diseases of the pregnant woman. Adverse pregnancy outcomes in pregnant women may be caused by a variety of factors, and researchers had found that urinary tract infections during pregnancy had also been linked to an increased risk of multiple adverse pregnancy outcomes [41–44]. Therefore, whether the adverse pregnancy outcomes were caused by sulfonamides exposure or infections during pregnancy aroused our attention. Two studies presented strategies for controlling protopathic bias, they used alternative antibiotics with the same indications and without adverse pregnancy outcomes as controls [16, 28], among which Muanda FT's research results showed that TMP-SMX exposure was associated with an increased risk of spontaneous abortion after controlling for indication bias and

protopathic bias. Control of indication bias was described in other studies, and the results show no significant association between observed adverse pregnancy outcomes and urinary tract infections and that confounding caused by urinary tract infection does not exist [45, 46]. We admit that this part of the controversy has never stopped, but we still want to emphasize that there might be a great risk of maternal exposure to sulfonamides during pregnancy.

Our study has several strengths. First of all, it is the first systematic analysis on the association between maternal exposure to sulfonamides and multiple adverse pregnancy outcomes. Our study included many high-quality articles with large study populations. We adopted the GRADE system to grade the evidence of the outcome indicators. Adverse pregnancy outcomes in our study included congenital malformations, spontaneous abortion, preterm birth/low birth weight as involved in both adversities in live births and fetal loss, and covered visible structural malformations (birth defects in organs and system) as well as "functional" manifestations (abortion, preterm birth/low birth weight), although abortion might be also due to potential malformations. Further detailed subgroup analyses were performed to clarify the impact of sulfonamides exposure during critical period of pregnancy on different types of congenital malformations.

Of course, there are some limitations in our study. First of all, few articles described the effect of sulfonamides exposure on a certain adverse pregnancy outcome, instead, multiple adverse pregnancy outcomes were often reported in one article. When we examined the effect of sulfonamides exposure during pregnancy on a specific adverse pregnancy outcome, data were limited and heterogeneity of combined outcomes was somehow difficult to control. We thus adopted the random effect model to merge the effect size, and carried out subgroup analysis and sensitivity analysis on the main results. Secondly, due to the limited information on the exact period when women were exposed to sulfonamides, we could not clearly distinguish the first, second or the third trimester of pregnancy. It also hindered us in identifying the possible key time windows that maternal sulfonamides exposure might affect adverse pregnancy outcomes. Thirdly, most of our data were extracted from maternal and infant database, prescription database, birth registrations and drug registries. Although the data were relatively complete and reliable, there still might be differences between the records and the actual medication intake of pregnant women. For instance, we were not sure whether women had taken the sulfonamides strictly as the way prescribed by doctors. Fourthly, some of the drugs we studied are sulfonamides alone, some are sulfonamides together with trimethoprim, which may cause some misclassifications of exposure group. We conducted a separate analysis to examine the effect of single sulfonamides use during pregnancy on children' s congenital malformations, and it showed that maternal exposure to sulfonamides alone during pregnancy was associated with congenital malformations (*OR* = 1.20, 95% *CI* 1.03–1.40) (S4 Fig). At the same time, we were unable to distinguish the effects of different sulfonamides and different doses on the pregnant outcome. In our recruited studies, one article reported the teratogenic effects of five different sulfonamides [19], such as sulfamethazine, sulfathiourea, sulfamethoxy-pyridazine, sulfamethoxydiazine and the combination of sulfamethazine-sulfathiourea-sulfa-methoxypyridazine. It argued that offspring' s ventricular septal defects were associated with sulfamethoxazine exposure during the second and third months of pregnancy. Clubfoot was associated with sulfathiourea use during the entire period of pregnancy as well as in the second and third months of pregnancy. Further studies are needed to confirm the impact of different kinds of sulfonamides during pregnancy and adverse pregnancy outcomes. Finally, there were also limited cohort studies and only one randomized controlled trial in the included literature, this is an important reason why the evidence level of GRADE is relatively low. We need more prospective research or experimental research that can provide a higher level of evidence to prove our research conclusions. The adverse pregnancy outcomes in this study were just

reported from the time of delivery till one year after birth. It was also reported that maternal exposure to antibiotics during pregnancy was associated with adverse outcomes in the postnatal and later developmental stages of the fetus, such as eczema, overweight and obesity in infant [47–49]. Long term effects of maternal exposure to sulfonamides are worth to be investigated, such as the physical and psychological development in childhood or even later life periods.

## Conclusion

In conclusion, maternal exposure to sulfonamides is possibly associated with offspring's congenital malformations. As infectious disease is harmful to both maternal and fetus health, meanwhile it's not easy to get the most useful drugs in time based on the possible or definite side effects, prescription of sulfonamides for pregnant women is suggested to be carefully censored.

## Supporting information

**S1 Checklist. PRISMA 2009 checklist.**
(DOC)

**S1 Table. Raw data.**
(XLSX)

**S1 Fig. Funnel plot for the association between maternal exposure to sulfonamides and congenital malformations.**
(TIF)

**S2 Fig. Funnel plot for the association between maternal exposure to sulfonamides and spontaneous abortion.**
(TIF)

**S3 Fig. Funnel plot for the association between maternal exposure to sulfonamides and preterm birth/low birth weight.**
(TIF)

**S4 Fig. Forest plot with pooled odds ratios of the effect of maternal exposure to single sulfonamides on offspring' s congenital malformations.**
(TIF)

## Author Contributions

**Conceptualization:** Peixuan Li, Xiaoyun Qin, Fangbiao Tao.

**Data curation:** Xiaoyun Qin, Kun Huang.

**Formal analysis:** Fangbiao Tao.

**Methodology:** Peixuan Li, Xiaoyun Qin, Kun Huang.

**Supervision:** Fangbiao Tao.

**Writing – original draft:** Peixuan Li.

**Writing – review & editing:** Fangbiao Tao, Kun Huang.

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
