## [Decision Letter · Decision Letter 0]

1 Sep 2020

PONE-D-20-21921

Maternal exposure to sulfonamides and adverse pregnancy outcomes: A systematic review and meta-analysis

PLOS ONE

Dear Dr. Huang,

Thank you for submitting your manuscript to PLOS ONE. After careful consideration, we feel that it has merit but does not fully meet PLOS ONE’s publication criteria as it currently stands. Therefore, we invite you to submit a revised version of the manuscript that addresses the points raised during the review process.

We look forward to receiving your revised manuscript.

Kind regards,

Linglin Xie

Academic Editor

PLOS ONE

Journal Requirements:

2. Please note that PLOS does not allow reference to data not shown (pages 11 and 16). Thus, before we proceed, we kindly ask you provide the relevant data within the manuscript, the Supporting Information files, or in a public repository. If the data are not a core part of the research study being presented, please remove any references to these data.

3. Please present the full electronic search strategy for at least one database, including any limits used, such that it could be repeated.

4. In addition to the results from Begg’s and Egger’s tests, please assess publication bias by graphical means, e.g funnel plot.

7. Please include your tables as part of your main manuscript and remove the individual files. Please note that supplementary tables (should remain/ be uploaded) as separate "supporting information" files.

Reviewers' comments:

Reviewer's Responses to Questions

**Comments to the Author**

1. Is the manuscript technically sound, and do the data support the conclusions?

Reviewer #1: Partly

Reviewer #2: Partly

2. Has the statistical analysis been performed appropriately and rigorously? 

Reviewer #1: Yes

Reviewer #2: I Don't Know

3. Have the authors made all data underlying the findings in their manuscript fully available?

Reviewer #1: Yes

Reviewer #2: Yes

4. Is the manuscript presented in an intelligible fashion and written in standard English?

Reviewer #1: No

Reviewer #2: Yes

5. Review Comments to the Author

Reviewer #1: Major comments:

1.Preterm birth usually is related to low birth weight, in this manuscript only preterm birth but not low birth weight was associated to maternal exposure to sulfonamides, its unusual. Author need to analysis the data again and make sure the results are reliable.

2.Infection disease is very harmful to both maternal and fetus, for pregnant women, its not easy to get the most useful drugs in time based on the possible or definite side effects. Authors of this manuscript side “some controversies about the safety of using sulfonamides during pregnancy”, but we can only see the author focused on the bad side of using sulfonamides during pregnancy, such as congenital malformations and preterm birth. Meanwhile, the OR for both congenital malformations and preterm birth are only a little high than 1 although. So, the conclusion of this article is a tendency or possible suggestion only.

Minor comments:

1. Some grammar mistake need to be corrected by a native English expert in the revised version.

Reviewer #2: 1. Please specify what time range were used for the article selection.

2. Please clarify the final sample size of the selected studies.

3. When evaluated the Maternal exposure to sulfonamides and preterm birth, no significant heterogeneity was found (I2=35.4%, P=0.213), why did the author still choose the random effects model?

4. When evaluated the Maternal exposure to sulfonamides and congenital malformations, what’s the rationale for choosing the two subgroups while the first trimester was involved in both group (“pre-pregnancy and first trimester” and “during the entire pregnancy”)?

5. In the discussion section, folate deficiency was considered to be one of the potential mechanisms for sulfonamide related adverse outcomes. Have the reviewed studies provide any information of folate levels?

6. Molloy, Anne M et al; (Food and nutrition bulletin 2008: S101-11; discussion S112-5) have reported the association between folate deficiency and lower birth weight, however, the authors find no significant relationship between sulfonamide exposure and lower birth weight. Please provide more explanation on this.

6. PLOS authors have the option to publish the peer review history of their article (what does this mean?). If published, this will include your full peer review and any attached files.

Reviewer #1: No

Reviewer #2: **Yes: **Zehuan Ding

---

## [Author Response · Author response to Decision Letter 0]

19 Oct 2020

PONE-D-20-21921

Maternal exposure to sulfonamides and adverse pregnancy outcomes: A systematic review and meta-analysis

PLOS ONE

Dear Dr. Xie,

On behalf of all co-authors, we thank you very much for giving us an opportunity to revise our manuscript, we appreciate editor and reviewers very much for their positive and constructive comments and suggestions on our manuscript entitled “Maternal exposure to sulfonamides and adverse pregnancy outcomes: A systematic review and meta-analysis”.

We have studied all the comments carefully and have made revision which marked in yellow in the paper. In the following pages are our point-by-point responses to each of the comments of the reviewers as well as your own comments.

We shall look forward to hearing from you at your earliest convenience.

Yours sincerely,

Kun Huang

Corresponding Author

E-mail Address: ahmuhuangk@163.com

Responds to Journal Requirements:

Response: We have carefully checked the entire manuscript for typography, format, and filename, and made sure that it met PLOS ONE’ s style requirements.

2. Please note that PLOS does not allow reference to data not shown (pages 11 and 16). Thus, before we proceed, we kindly ask you provide the relevant data within the manuscript, the Supporting Information files, or in a public repository. If the data are not a core part of the research study being presented, please remove any references to these data.

Response: We have provided the relevant data instead of “data not shown” (pages 11 and 16) as the Fig 6 and the S4 Fig. In Fig 6, we have combined preterm birth and low birth weight as one outcome based on the reviewer’ s comments and re-made the forest plot.

3. Please present the full electronic search strategy for at least one database, including any limits used, such that it could be repeated.

Response: We have presented the full electronic search strategy that had been used as the following retrieval formula in Pubmed: (pregnancy OR pregnant OR conception OR fetation OR gestation) AND (sulfonamides OR sulfamethoxazole OR sulfametoxydiazine OR sulfaquinoxaline OR sulfamethazine OR sulfadiazine OR sulfasalazine OR sulfamethizole OR cotrimoxazole). There are no other special restrictions.

4. In addition to the results from Begg’s and Egger’s tests, please assess publication bias by graphical means, e.g funnel plot.

Response: We completely agree with this valuable suggestion. We added funnel plot as a supplementary information (S1 S2 and S3 Fig) to further assess the judgment of publication bias.

Response: We have added the raw data as the excel file named “raw data”.

Response: We have provided the relevant data instead of “data not shown” (pages 11 and 16) as the Fig 6 the S4 Fig. In Fig 6, we have combined preterm birth and low birth weight as one outcome based on the reviewer’ s comments and re-made the forest plot.

7.Please include your tables as part of your main manuscript and remove the individual files. Please note that supplementary tables (should remain/ be uploaded) as separate "supporting information" files.

Response: We have added our tables to the main manuscript and uploaded the supplementary table (excel file) as separate “Supporting information” file.

Response: We have added captions for Supporting Information files at the end of the manuscript, and updated in-text citations to match accordingly. 

Responds to the reviewer’s comments:

Reviewer #1: Major comments:

1.Preterm birth usually is related to low birth weight, in this manuscript only preterm birth but not low birth weight was associated to maternal exposure to sulfonamides, its unusual. Author need to analysis the data again and make sure the results are reliable.

Response: Thank you for the reviewer’ s comments. Indeed, preterm birth is usually related to low birth weight. While in some cases, some full-term infants are low birth weight, and some preterm infants have normal birth weights. That’ s why in Ratanajamit C’ s study, the authors had clearly indicated that low birth weight was restricted to full-term deliveries. In our meta-analysis, only three included articles concerning maternal exposure to sulfonamides and preterm birth/low birth weight (including Ratanajamit C’ s study). In Yang J’ s study, low birth weights covered preterm birth, and vice versa. Prasad MH’ s study just included preterm birth without mentioning low birth weight. 

We fully considered the reviewer’ s comments and made a new analysis by combining preterm birth and low birth weight as one outcome. The findings revealed that there was no statistically significant effect of maternal exposure to sulfonamides on preterm birth/low birth weight (OR=1.38, 95% CI 0.94-2.03). We have also added some words in the discussion (Page 15-16). We believe that more research data is needed to identify the relationship between maternal exposure of sulfonamides and preterm birth/low birth weight.

2.Infection disease is very harmful to both maternal and fetus, for pregnant women, its not easy to get the most useful drugs in time based on the possible or definite side effects. Authors of this manuscript side “some controversies about the safety of using sulfonamides during pregnancy”, but we can only see the author focused on the bad side of using sulfonamides during pregnancy, such as congenital malformations and preterm birth. Meanwhile, the OR for both congenital malformations and preterm birth are only a little high than 1 although. So, the conclusion of this article is a tendency or possible suggestion only.

Response: Thank you for your valuable comments. We fully agree that it’ s not easy to get the most useful drugs in time based on the possible or definite side effects for pregnant women, although Muanda FT. et al. had shown that sulfonamides were associated with adverse pregnancy outcomes after controlling for indication bias and protopathic bias. We have taken the reality into account and rephrased the words in the discussion (Page 15-20) and conclusion (Page 20). We admit that this part of the controversy has never stopped and then interpret the conclusion more prudently.

3.Minor comments:

1. Some grammar mistake need to be corrected by a native English expert in the revised version.

Response: We apologize for the mistakes in the manuscript. We have carefully checked, corrected and polished the entire manuscript.

Reviewer #2: 

1.Please specify what time range were used for the article selection.

Response: In the initial literature search, we had limited the time range as the recent ten and fifteen years and there were few related articles. The time rage was then not used in the final article selection.

2.Please clarify the final sample size of the selected studies.

Response: We have listed the individual sample size for each study in Table 1. According to the reviewer’ s comments, we have added up all the numbers of case and the final sample size is 1,096,350. We have also added this number in the text (Page 10). 

3.When evaluated the Maternal exposure to sulfonamides and preterm birth, no significant heterogeneity was found (I2=35.4%, P=0.213), why did the author still choose the random effects model?

Response: Thank you for the reviewer’ s comments. In the revised manuscript, we have combined preterm birth and low birth weight as one outcome, and used the random effects model (Page 12-13) in the analysis.

4.When evaluated the Maternal exposure to sulfonamides and congenital malformations, what’s the rationale for choosing the two subgroups while the first trimester was involved in both group (“pre-pregnancy and first trimester” and “during the entire pregnancy”)?

Response: Thank you for the reviewer’ s comments. It’ s also what had troubled us during the analysis. We tried to divide the group of maternal sulfonamides into pre-pregnancy, the first, second and third trimester of pregnancy in initial analysis plan. After careful examining the included articles, we found that it did not meet the grouping criteria. Most articles used “pre-pregnancy and first trimester” as an independent exposure period, and no study clearly distinguished the second, third trimester of pregnancy. So we could not clearly divide the three trimesters of pregnancy, and had adopted “pre-pregnancy and first trimester” and “during the entire pregnancy” based on the existed grouping methods in the included articles. Indeed, the first trimester was involved in both groups. It’s one of the limitations in the paper, and we had added some words in the discussion section (Page 18-19).

The period of pre-pregnancy and first trimester is the key time window for folate supplementation to prevent fetal malformations, and sulfonamides may affect folic acid synthesis and metabolism.

5.In the discussion section, folate deficiency was considered to be one of the potential mechanisms for sulfonamide related adverse outcomes. Have the reviewed studies provide any information of folate levels?

Response: We have carefully checked the original literatures we included. It is a pity that they did not provide information on folate levels. Most of the included articles explained information on folate antagonism in terms of mechanisms. We detailed the mechanism of sulfonamides related to folate deficiency in the discussion.

6.Molloy, Anne M et al; (Food and nutrition bulletin 2008: S101-11; discussion S112-5) have reported the association between folate deficiency and lower birth weight, however, the authors find no significant relationship between sulfonamide exposure and lower birth weight. Please provide more explanation on this.

Response: Thank you for the reviewer’ s comments. Folate deficiency might be one of the potential metabolisms that sulfonamides may lead to adverse pregnancy outcome. As we have understood, it does not mean that maternal sulfonamides exposure will necessarily cause folate deficiency, and that folate deficiency will necessarily lead to low birth weight. As shown in Ratanajamit C’ s study, there was no significant correlation between maternal exposure to sulfonamides and low birth weight. And also, just as the reviewer had pointed out, we did not have information on folate levels in the included articles, which made it difficult to clearly identify the relationship among sulfonamides exposure. folate deficiency and adverse pregnant outcomes.

Preterm birth is usually related to low birth weight. In our meta-analysis, only three included articles concerning maternal exposure to sulfonamides and preterm birth/low birth weight. Ratanajamit C’ s study had clearly indicated that low birth weight was restricted to full-term deliveries. While in Yang J’ s study, low birth weights covered preterm birth, and vice versa. And Prasad MH’ s study just included preterm birth without mentioning low birth weight. In the revised version, we have made a new analysis by combining preterm birth and low birth weight as one outcome. The findings revealed that there was no statistically significant effect of maternal exposure to sulfonamides on preterm birth/low birth weight (OR=1.38, 95% CI 0.94-2.03). We have also added some words in the discussion (Page 15-16). We believe that more research data is needed to identify the relationship between maternal exposure of sulfonamides and preterm birth/low birth weight.

---

## [Decision Letter · Decision Letter 1]

4 Nov 2020

Maternal exposure to sulfonamides and adverse pregnancy outcomes: A systematic review and meta-analysis

PONE-D-20-21921R1

Dear Dr. Huang,

We’re pleased to inform you that your manuscript has been judged scientifically suitable for publication and will be formally accepted for publication once it meets all outstanding technical requirements.

Kind regards,

Linglin Xie

Academic Editor

PLOS ONE

Additional Editor Comments (optional):

Reviewers' comments:

Reviewer's Responses to Questions

**Comments to the Author**

1. If the authors have adequately addressed your comments raised in a previous round of review and you feel that this manuscript is now acceptable for publication, you may indicate that here to bypass the “Comments to the Author” section, enter your conflict of interest statement in the “Confidential to Editor” section, and submit your "Accept" recommendation.

Reviewer #1: All comments have been addressed

Reviewer #2: All comments have been addressed

2. Is the manuscript technically sound, and do the data support the conclusions?

Reviewer #1: Yes

Reviewer #2: Yes

3. Has the statistical analysis been performed appropriately and rigorously? 

Reviewer #1: Yes

Reviewer #2: I Don't Know

4. Have the authors made all data underlying the findings in their manuscript fully available?

Reviewer #1: Yes

Reviewer #2: Yes

5. Is the manuscript presented in an intelligible fashion and written in standard English?

Reviewer #1: Yes

Reviewer #2: Yes

6. Review Comments to the Author

Reviewer #1: (No Response)

Reviewer #2: (No Response)

7. PLOS authors have the option to publish the peer review history of their article (what does this mean?). If published, this will include your full peer review and any attached files.

Reviewer #1: No

Reviewer #2: No

---

## [Editor Report · Acceptance letter]

13 Nov 2020

PONE-D-20-21921R1 

Maternal exposure to sulfonamides and adverse pregnancy outcomes: A systematic review and meta-analysis 

Dear Dr. Huang:

I'm pleased to inform you that your manuscript has been deemed suitable for publication in PLOS ONE. Congratulations! Your manuscript is now with our production department. 

Kind regards, 

on behalf of

Dr. Linglin Xie 

Academic Editor

PLOS ONE